# Ultrafast pseudospin quantum beats in multilayer WSe$_2$ and MoSe$_2$

**Simon Raiber[1], Paulo E. Faria Junior[2], Dennis Falter[1], Simon Feldl[1], Petter Marzena[1], Kenji Watanabe** [3]**, Takashi Taniguchi** [4]**, Jaroslav Fabian** [2] **& Christian Schüller** [1] ✉

Layered van-der-Waals materials with hexagonal symmetry offer an extra degree of freedom to their electrons, the so-called valley index or valley pseudospin, which behaves conceptually like the electron spin. Here, we present investigations of excitonic transitions in mono- and multilayer WSe$_2$ and MoSe$_2$ materials by time-resolved Faraday ellipticity (TRFE) with in-plane magnetic fields, $B_\parallel$, of up to 9 T. In monolayer samples, the measured TRFE time traces are almost independent of $B_\parallel$, which confirms a close to zero in-plane exciton $g$ factor $g_\parallel$, consistent with first-principles calculations. In contrast, we observe pronounced temporal oscillations in multilayer samples for $B_\parallel > 0$. Our first-principles calculations confirm the presence of a non-zero $g_\parallel$ for the multilayer samples. We propose that the oscillatory TRFE signal in the multilayer samples is caused by pseudospin quantum beats of excitons, which is a manifestation of spin- and pseudospin layer locking in the multilayer samples.

The semiconducting transition-metal dichalcogenides (TMDCs) hold great promise for optoelectronic applications, since they form direct bandgap semiconductors in the monolayer limit. Their optical properties are governed by excitons, i.e., Coulomb-bound electron-hole pairs[1,2], even at room temperature, due to extraordinarily large exciton binding energies. For high-quality encapsulated MoSe$_2$ monolayers, superior optical quality with exciton linewidths approaching the lifetime limit has been demonstrated[3,4]. Furthermore, anomalous, non-classical diffusion behavior of excitons has been detected[5,6] and calculated[7] for TMDC monolayers. Starting from bilayers, the bandgap becomes indirect. Nevertheless, going from a single layer to multilayers, the direct interband transitions at the K points of the Brillouin zone still dominate the optical absorption[8]. Another property of monolayer material is the strong spin-orbit coupling in combination with inversion asymmetry, which lead to large valley-selective spin-orbit splittings of the band edges, culminating in the so-called spin-valley locking. This peculiarity is appreciated by the introduction of a pseudospin index, which conceptually behaves like the electron spin,

and is connected to the occupation of the two non-equivalent K$^+$ and K$^-$ valleys of the first Brillouin zone. Interestingly, the spin-valley locking of a single layer transforms into a spin- or pseudospin-layer locking for multilayers[9]. For TMDC bilayers it has even been suggested that the spin-layer locking can be exploited for the design of spin quantum gates[10].

Interlayer excitons (IX), where electron and hole reside in adjacent layers, were first detected in heterobilayers[11]. There, the characteristics of IX depend crucially on the material combination[12–14]. Recently, even valley-polarized currents of IX in heterobilayers have been demonstrated[15]. While in heterobilayers the oscillator strength of IX is weak, the situation can be different for homobilayers or multilayers[16]. In MoS$_2$ bilayers, strong absorption by IX up to room temperature was reported[17–22]. In MoSe$_2$, the situation is similar to MoS$_2$, though the oscillator strength of the IX is smaller[23]. Nevertheless, IX have been reported in H-stacked MoTe$_2$[24] and MoSe$_2$[23,25] multilayers. In contrast to Mo-based multilayers, the momentum-space direct IX in W-based materials has so far not been observed. It should be noted that for

[1]Institut für Experimentelle und Angewandte Physik, Universität Regensburg, D-93040 Regensburg, Germany. [2]Institut für Theoretische Physik, Universität Regensburg, D-93040 Regensburg, Germany. [3]Research Center for Functional Materials, National Institute for Materials Science, Tsukuba, Ibaraki 305-0044, Japan. [4]International Center for Materials Nanoarchitectonics, National Institute for Materials Science, Tsukuba, Ibaraki 305-0044, Japan. ✉e-mail: christian.schueller@ur.de

WSe$_2$ homobilayers, IX due to momentum-indirect transitions below the optical bandgap were reported[26,27].

While monolayer TMDCs have been quite intensely investigated in out-of-plane magnetic fields, investigations on multilayer samples are quite rare. The out-of-plane $g$ factor, $g_\perp$, of the intralayer A excitons is in MoSe$_2$ and WSe$_2$ multilayers smaller in magnitude than in single layers[25,28]. So far, there are, however, no experimental investigations on the in-plane $g$ factor, $g_\parallel$, in TMDC multilayers available. In-plane magnetic fields, $B_\parallel$, have been applied to TMDC monolayers for the brightening of dark excitonic states via mixing of the spin levels by the in-plane field[29–32]. In this work, we present time-resolved Faraday ellipticity (TRFE) experiments on WSe$_2$ and MoSe$_2$ mono- and multi-layers in in-plane magnetic fields. While we do not observe a significant influence of in-plane fields of up to 9 T in experiments on monolayers, pronounced temporal oscillations are observed in the TRFE time traces of multilayers for $B_\parallel > 0$. Remarkably, the derived in-plane exciton $g$ factors, $|g_\parallel|$, are close to reported $|g_\perp|$ values of the same materials[25].

## Results and discussion

### Sample characterization

We start the discussion with reflectance-contrast (RC) experiments of the investigated samples, in order to characterize the excitonic transitions in the materials. Simplified schematic drawings of the first two A excitons, A$_{1s}$ and A$_{2s}$, in a multilayer sample are plotted in Fig. 1e. Figure 1a shows an overview of RC spectra of the four samples, investigated in the main body of the manuscript. Excitonic transitions

are marked by small vertical arrows as derived from fitting the RC spectra with a transfer-matrix model, assuming complex Lorentz oscillators for the excitonic transitions (see Supplementary Information). The schematic drawings in the inset of Fig. 1a depict the samples, which are MoSe$_2$ and WSe$_2$ mono- and multilayers (for more details, see the methods section). The monolayer samples are encapsulated in hBN to protect them from environmental influences and to provide a homogeneous dielectric environment. The WSe$_2$ multilayer constits of about 14 layers, the MoSe$_2$ multilayer is much thicker, counting about 84 layers, as determined by atomic-force microscopy. In both monolayer samples, the intralayer A$_{1s}$ excitons show up as distinct and sharp features in the RC spectra in Fig. 1a. In the MoSe$_2$ monolayer also the B$_{1s}$ exciton can be detected, while for the WSe$_2$ monolayer it is outside the displayed energy range. For clarity, the transitions at the K points of the first Brillouin zone, which lead to the excitonic A and B resonances are sketched for both materials in Fig. 1b, c (see layer 1, only, for the monolayer case). Interlayer transitions are omitted in the schematic pictures, since they play no role in our experiments. Going from the monolayer to multilayers, the intralayer excitonic resonances show a redshift, and the energetic separation between A$_{1s}$ and A$_{2s}$ decreases because of the stronger dielectric screening[24,25]. In agreement with published results[25], we observe in Fig. 1a in the WSe$_2$ multilayer two features, which can be attributed to the A$_{1s}$ and A$_{2s}$ intralayer excitons. This assignment is supported by our excitonic calculations (see methods section): From the derived effective masses for electron and hole we calculate an energetic separation of the A$_{1s}$

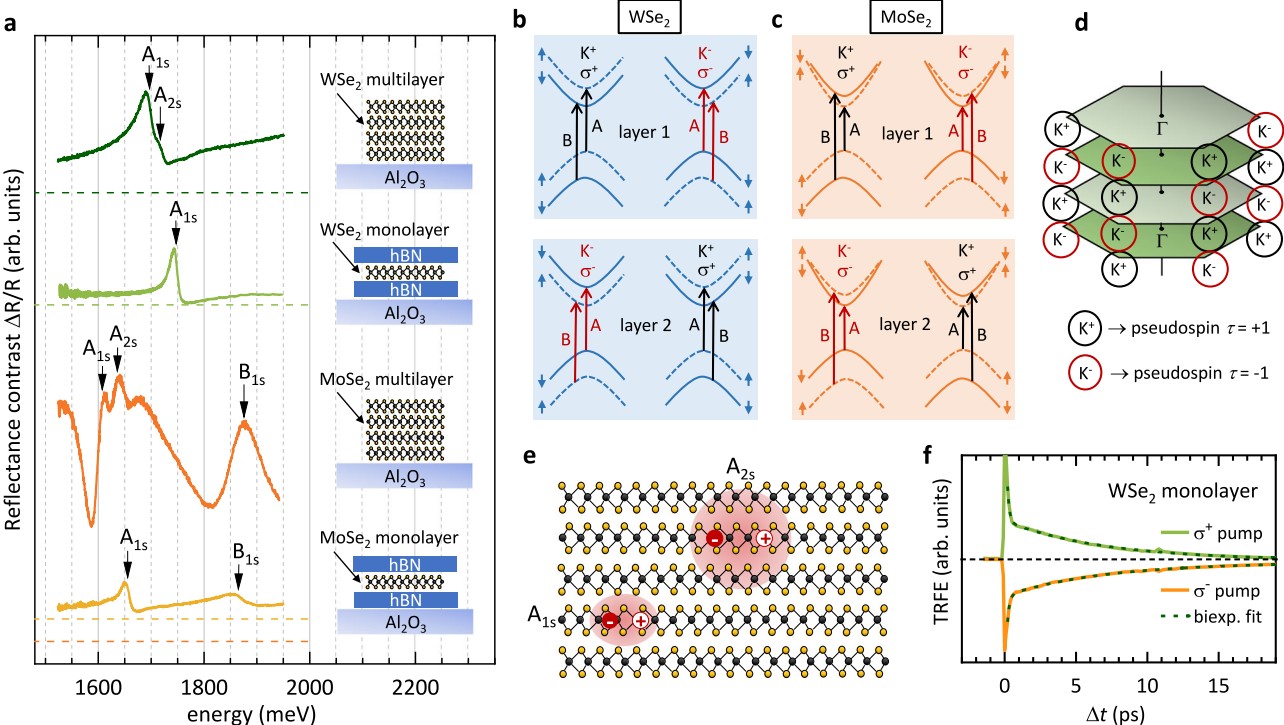

**Fig. 1 | Reflectance-contrast and TRFE experiments, intralayer transitions, and pseudospin-layer locking. a** White-light reflectance-contrast experiments of the investigated samples: MoSe$_2$ and WSe$_2$ monolayers, encapsulated in hBN, and multilayer samples of both materials. The corresponding zero lines are given as dashed lines of the same color. All samples are prepared on transparent sapphire substrates. The substrate temperature in all RC measurements was $T \sim 20$ K, except for the MoSe$_2$ multilayer, where it was $< 10$ K, as derived from the intensity ratio of ruby lines from the substrate (see methods section). Excitonic transitions, as derived from a transfer-matrix-model fit, are indicated by small vertical arrows. We note that the TRFE experiments are all performed at $T \sim 5$ K. **b** Schematic picture of

momentum- and spin-allowed transitions in an H-type WSe$_2$ bilayer. **c** Same as **b** but for an MoSe$_2$ bilayer. **d** Schematic picture of the layer Brillouin zones in a four-layer structure. Due to the 180° rotation between neighboring layers in an H-type structure, K$^+$ and K$^-$ valleys are alternating. An interlayer pseudospin $\tau = +1$ is connected to the K$^+$ valleys of the individual layers (marked by black circles), while $\tau = -1$ corresponds to the K$^-$ valleys (indicated by red circles). **e** Sketch of intralayer excitons A$_{1s}$ and A$_{2s}$. **f** TRFE traces of the encapsulated WSe$_2$ monolayer for $\sigma^+$ (green solid line) and $\sigma^-$ (orange solid line) pump helicities, measured in resonance with the A$_{1s}$ exciton. The dashed lines are biexponential fits to the data.

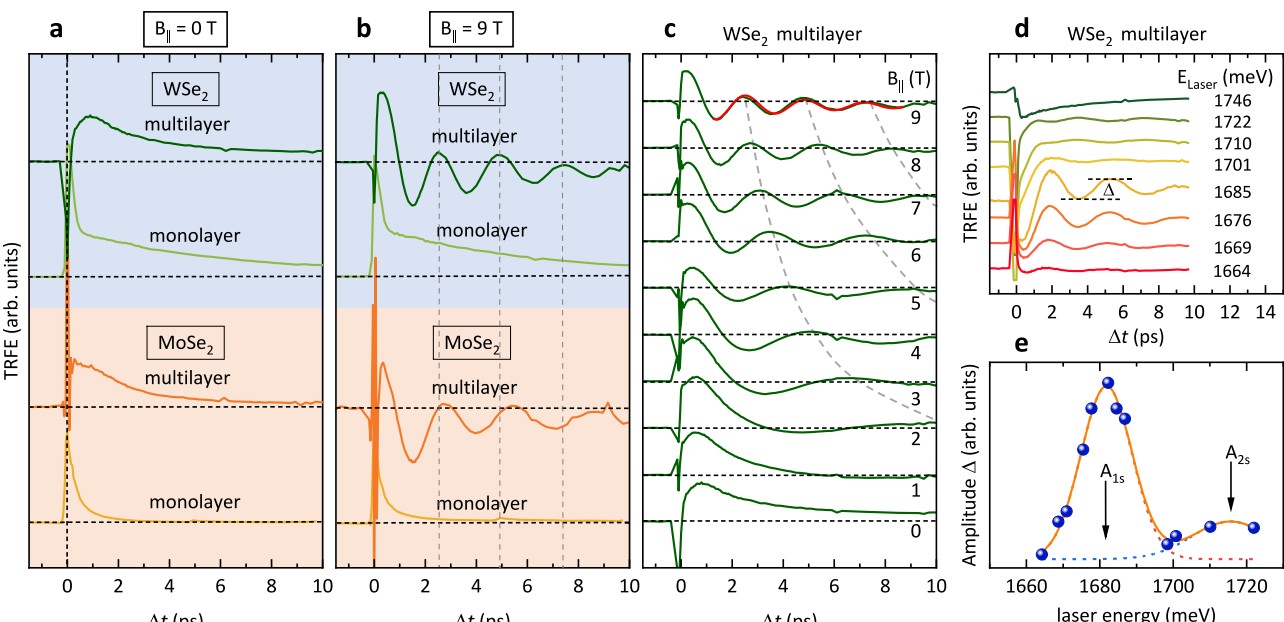

**Fig. 2 | TRFE experiments in in-plane magnetic fields.** Comparison of TRFE traces of all samples at **a** $B_\parallel = 0$, and, **b** $B_\parallel = 9$ T, excited at the $A_{1s}$ excitonic resonances. In the multilayer samples, strong temporal oscillations are observed in the time traces at $B_\parallel = 9$ T, in contrast to the monolayer samples, which show no oscillations. **c** TRFE traces of the $WSe_2$ multilayer for different in-plane magnetic fields. The red solid line represents an exponentially-damped cosine fit to the data. The dashed gray lines are guides to the eye. **d** TRFE measurements of the $WSe_2$ multilayer at fixed in-plane field $B_\parallel = 6$ T for different center energies $E_{Laser}$ of the laser pulses, as given in the figure. A clear resonance behavior of the signal can be observed. The signal amplitude $\Delta$, as determined for all curves, is indicated. **e** Plot of the extracted signal amplitudes $\Delta$ (blue solid bullets), as indicated in **d**, versus central laser energy. The dashed lines represent Gaussian fits, while the solid orange line is the sum of both fit curves. Resonances with the $A_{1s}$ and $A_{2s}$ excitons are indicated by arrows. For all measurements the temperature was $T \sim 5$ K.

and $A_{2s}$ intralayer excitons of ~20.8 meV, which is very close to the experimental value of ~19.2 meV. In agreement with the reports in ref. 25, we also do not find a feature, related to the IX in $WSe_2$ multilayer in our RC experiments in Fig. 1a. Also for the $MoSe_2$ multilayer, we do not observe a spectral feature, related to the IX. Similar to ref. 33, we find spectral features related to the $A_{1s}$ and $A_{2s}$ intralayer excitons in the $MoSe_2$ multilayer. Again, this assignment in Fig. 1a is corroborated by our computed energy separation of $A_{1s}$ and $A_{2s}$ excitons of ~31.8 meV, which is close to the experimental value of ~28.6 meV.

Figure 1 d is a sketch of the individual first Brillouin zones of an H-type four-layer structure. In H-type structure, subsequent layers are rotated by 180°. Therefore, in momentum space, $K^+$ and $K^-$ valleys of the individual layers are alternating, which is called spin-layer locking[9,10]. A pseudospin quantum number $\tau = +1(-1)$ can be attributed to the $K^+$ ($K^-$) valley, leading to a pseudospin-layer locking.

Figure 1 f shows typical TRFE time traces, recorded on the $WSe_2$ monolayer at zero magnetic field under resonant excitation of the $A_{1s}$ exciton. All experiments presented in this manuscript are in the excitonic regime, i.e., the exciton densities are below the Mott density (see methods section). The light green line shows a trace with $\sigma^+$-polarized pump pulses, which create a $K^+$ valley polarization at time $\Delta t = 0$. The orange line is an analogous measurement but with $\sigma^-$ pump pulses, i.e., a $K^-$ valley polarization is initialized. The dashed lines represent biexponential fits to the data. Both measurement curves can be nicely fitted by a biexponential decay with a short time constant of $\tau_r \sim 0.15$ ps and a longer decay time of $\tau_v \sim 7.0$ ps. There are a couple of different processes, which can contribute to the fast decay at short times. Among them is the direct radiative decay of excitons, which are created inside the light cone, and which directly decay radiatively before any scattering event can take place. Our measured $\tau_r$ of ~0.15 ps is in very good agreement with previous measurements of the radiative lifetime of excitons in $WSe_2$ monolayers[34]. Therefore, it is likely that the fast initial decay of the TRFE signal is influenced by direct radiative recombination of part of the exciton population, created inside the light cone. A

significant part of the excitonic population is, however, scattered out of the light cone, e.g., by phonons, and contributes to the valley polarization over a longer time period. We note that also exciton localization in traps and diffusion, as, e.g., observed for semiconductor nanoplatelets[35], may contribute to a prolonged exciton lifetime. The main mechanism leading to valley relaxation in $WSe_2$ monolayers, is the long-range exchange mechanism between electron and hole, which is proportional to the center-of-mass momentum of the exciton[36–38]. The valley-polarization decay time of $\tau_v \sim 7.0$ ps, extracted from the TRFE traces of the hBN encapsulated $WSe_2$ monolayer in Fig. 1f, is in very good agreement with the reported decay time of 6.0 ps, measured on a bare $WSe_2$ monolayer on a $SiO_2$ substrate in ref. 36, and with calculations, based on the long-range exchange mechanism[37].

## TRFE experiments in in-plane magnetic fields

We now move on to the central point of the investigations in this manuscript: experiments in in-plane magnetic fields, $B_\parallel$. Figure 2a shows a comparison of TRFE traces of all four investigated samples at $B_\parallel = 0$, where the laser was tuned in resonance with the $A_{1s}$ excitonic resonances in the respective materials, as marked by arrows in the RC measurements in Fig. 1a. The trace of the $WSe_2$ monolayer is the same as shown in Fig. 1f ($\sigma^+$ pump). Comparing the two monolayer samples in Fig. 2a, one can recognize the much faster valley depolarization in $MoSe_2$. The measured decay time is here ~1 ps, as compared to ~7 ps for the $WSe_2$ monolayer (see discussion above). The much faster valley depolarization in the $MoSe_2$ monolayer is reminiscent of a close to zero valley polarization, measured in cw polarized photoluminescence on this material[39]. Surprisingly, while the valley depolarization time is comparable for the $WSe_2$ monolayer and multilayer, it is much longer in the $MoSe_2$ multilayer, as compared to the monolayer. This may be related to the fact that in $MoSe_2$ monolayers the lowest energy state is a bright state, which is different in all other samples, however, we note here that this is not the focus of this work. In Fig. 2b, the same measurements are shown, now for an in-plane field of $B_\parallel = 9$ T. While the

TRFE time traces for the monolayer samples are essentially unchanged when compared to $B_\parallel = 0$, they are significantly different for the multilayer samples. Strong and pronounced oscillations can be observed. The oscillation period of the MoSe$_2$ multilayer is slightly longer than for the WSe$_2$ multilayer. As a guide to the eye, vertical dashed lines are plotted in Fig. 2b, which mark the maxima of the oscillations of the WSe$_2$ multilayer. Figure 2c shows a full data set for the WSe$_2$ multilayer from $B_\parallel = 0$ T to 9 T. A full data set of the MoSe$_2$ multilayer is plotted in the supplementary Fig. S2. The gray dashed lines in Fig. 2c are guides to the eye and mark the oscillation maxima, which correspond to the same oscillation period. To test the resonance behavior of the TRFE measurements, we plot in Fig. 2d TRFE traces of the WSe$_2$ multilayer at fixed in-plane field of $B_\parallel = 6$ T for different central energies of the laser pulses. The central energies are given in Fig. 2d, the spectral widths of the pulses is ~ 16 meV. We extract the amplitudes of the oscillations, $\Delta$, as indicated in Fig. 2d, and plot them versus central laser energy in Fig. 2e. The amplitudes show a clear resonance behavior. The dashed lines in Fig. 2e are Gaussian fits, and the solid orange line is the sum of the two Gaussian fit curves. The two maxima can be attributed to resonances with the A$_{1s}$ and A$_{2s}$ intralayer excitons (cf. with the resonance features in the RC experiments in Fig. 1a). We note that the A$_{1s}$ resonance position is shifted by about 16 meV to lower energies in comparison to the white-light RC measurements in Fig. 1a, which can be due to bandgap-renormalization effects[40] and/or a temperature increase under pulsed excitation. If the redshift would be caused entirely by a temperature increase, the temperature in the TRFE experiments on the WSe$_2$ multilayer could be up to $T \sim 100$ K[41] as an upper limit. A full dataset of TRFE traces in resonance with the A$_{2s}$ exciton from 0 to 9 T can be found in supplementary Fig. S3 (same for the MoSe$_2$ multilayer in Fig. S4). It should be emphasized that we do not observe oscillations, i.e., an excitonic resonance, at energies above the A$_{1s}$ and A$_{2s}$ excitons in the MoSe$_2$ multilayer, in the spectral region, where in ref. 25 an IX was reported in RC measurements. From that we conclude that for our observed temporal oscillations only the intralayer A excitons are relevant.

Clearly, the oscillations in the TRFE traces resemble coherent precession of a magnetic moment about the in-plane magnetic field, as known from, e.g., electron spins in n-doped GaAs bulk[42], hole spins in GaAs quantum wells[43], or, localized background charge carriers in MoS$_2$ and WS$_2$[44], among many other examples. We have fitted all experimental curves for $B_\parallel > 0$ with an exponentially-damped cosine function $S(\nu, \tau_\nu) \propto \exp(-\Delta t / \tau_\nu) \cos(2\pi\nu\Delta t)$ for delay times $\Delta t$ well above the fast initial decay of the TRFE signals, as exemplarily shown by the red solid line in Fig. 2c for the 9 T trace. An important result is that the oscillations with frequency $\nu$ at $B_\parallel > 0$ decay with approximately the same decay time $\tau_\nu$ as the excitonic signal at $B_\parallel = 0$, and no long-lived oscillatory signal is developed. From that we conclude that the oscillations stem from a Larmor precession of the exciton magnetic moment, and not from the spin of background charge carriers, as observed for localized electrons in MoS$_2$ and WS$_2$ monolayers[44]. Furthermore, the approximate independence of the decay time $\tau_\nu$ from $B_\parallel$ shows that $g$ factor fluctuations do not play a role. Otherwise, a $1/B_\parallel$ dependence of $\tau_\nu$ would be expected[45,46]. Figure 3a shows a summary of all oscillation frequencies $\nu$, extracted by this procedure, versus $B_\parallel$. Clearly, a linear, Zeeman-like dependence can be recognized. The determined $|g_\parallel|$ are given in the legend of Fig. 3a. The experimental error margins for these values are about ±0.2. It should be noted that with TRFE experiments we can only determine the magnitude of the $g$ factor but not its sign. Very remarkably, for all excitonic resonances, the determined $|g_\parallel|$ are very close to out-of-plane $g$ factors, $|g_\perp|$, of the corresponding materials, reorted in refs. 25, 28, which are for WSe$_2$ bulk material $|g_\perp| = 3.2 \pm 0.2$ and $3.3 \pm 0.6$ for the A$_{1s}$ and A$_{2s}$ intralayer excitons, respectively[25]. For MoSe$_2$ bulk, the reported value for A$_{1s}$ is $|g_\perp| = 2.7 \pm 0.1$[25]. Hence, we conclude that $|g_\parallel| \sim |g_\perp|$ for multilayer TMDCs, approaching the bulk limit.

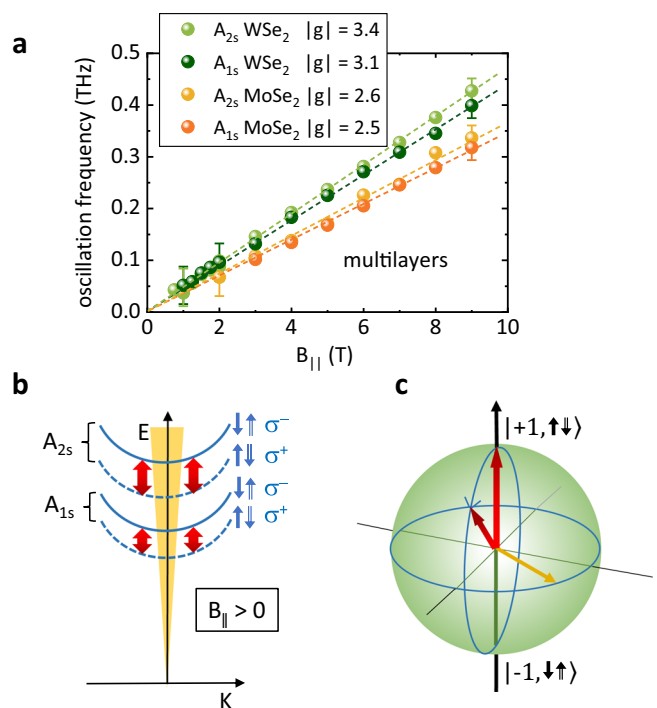

**Fig. 3 | Extracted in-plane $g$ factors, and proposed mechanism. a** Experimentally detected oscillation frequencies for A$_{1s}$ and A$_{2s}$ intralayer excitons (orange and yellow solid bullets) in MoSe$_2$, and, A$_{1s}$ and A$_{2s}$ intralayer excitons (dark green and light green solid bullets) in WSe$_2$ multilayer samples versus in-plane magnetic field. Exemplary error bars for low and high fields are indicated. The extracted absolute values $|g_\parallel|$ are given in the legend. The experimental error margins are about ± 0.2. **b** Energy versus center-of-mass momentum $K$ dispersion of intralayer A excitons at $B_\parallel > 0$ in a multilayer. For the excitons, the $z$ component of the spin of the electron is symbolized by a small arrow with a single line, while the hole spin is indicated by an arrow with a double line. Energy splittings of the excitons due to finite $g_\parallel$ are taken into account. The red double arrows should symbolize the coherent momentum-allowed oscillation between adjacent layers. **c** Representation of the pseudospin rotation on a Bloch sphere. The north pole corresponds to $\tau = +1$, while the south pole represents the $\tau = -1$ state. The orange arrow would correspond to a system, excited with linearly-polarized light.

## Comparison to first-principles calculations and discussion

In the following we will discuss our experimental findings further and compare them to first-principles calculations (see methods section). In Table 1, computed spin- and orbital angular momenta for out-of-plane ($S_z^i, L_z^i$) as well as in-plane ($S_x^i, L_x^i$) directions are given for the monolayer and multilayer samples. The superscript i stands for CB or VB, i.e., for the conduction-band or valence-band states, respectively, which are relevant for the intralayer A excitons in the materials (cf. Fig. 1b, c). We note that the relevant CB states are different for the two materials because of the reverse spin order. The computed $g$ factors for the A excitons, which are determined by $g_{\perp/\parallel} = 2(S_{z/x}^{CB} + L_{z/x}^{CB} - S_{z/x}^{VB} - L_{z/x}^{VB})$[47] are also given. The minus signs in front of the VB angular momenta account for the fact that the angular momentum of a hole is just opposite to the angular momentum of an electron in the VB state. One can see that for the monolayers and WSe$_2$ multilayers, the calculated $g_\perp$ agree well with published experimental values, while for MoSe$_2$ multilayers the computed $|g_\perp|$ is somewhat smaller as compared to the experimental report. The experimental result of $g_\parallel \sim 0$ for the monolayers is confirmed by the calculations, which give exactly $g_\parallel = 0$ (both, spin and orbital angular momenta contributions are zero due to symmetry considerations and verified numerically, cf. table 1).

For the multilayer samples on the other hand, the calculations do deliver nonzero $g_\parallel$, consistent with our experimental finding, though

**Table 1 | Computed values of out-of-plane and in-plane spin-, S, and orbital, L, angular momenta for the conduction-band (CB) and valence-band (VB) states, which are relevant for the A excitons of the investigated materials**

| Material | $S_z^{CB}$ | $L_z^{CB}$ | $S_z^{VB}$ | $L_z^{VB}$ | $g_\perp$ | $g_\perp$ (exp.) | $S_x^{CB}$ | $L_x^{CB}$ | $S_x^{VB}$ | $L_x^{VB}$ | $\lvert g_\parallel \rvert$ | $\lvert g_\parallel \rvert$ (exp.) |
|---|---|---|---|---|---|---|---|---|---|---|---|---|
| WSe$_2$ monolayer | 0.98 | 2.97 | 1.00 | 5.00 | −4.10 | −4.38...−1.57[a] | 0.00 | 0.00 | 0.00 | 0.00 | 0.00 | ~0 |
| WSe$_2$ multilayer | 0.97 | 2.98 | 1.00 | 4.40 | −2.89 | −3.4...−2.3[b] | 0.00 | 0.00 | 0.47 | ±0.07 | 0.80...1.08 | 3.1 ± 0.2 |
| MoSe$_2$ monolayer | 1.00 | 1.81 | 1.00 | 3.96 | −4.30 | −4.4...−3.8[c] | 0.00 | 0.00 | 0.00 | 0.00 | 0.00 | ~0 |
| MoSe$_2$ multilayer | 1.00 | 1.76 | 1.00 | 2.67 | −1.84 | −2.7[d] | 0.00 | 0.00 | 0.74 | ±0.06 | 1.36...1.60 | 2.5 ± 0.2 |

For the first-principles calculations, see the methods section. The corresponding theoretical g factors, $g_\perp$ and $g_\parallel$, for the A excitons are given. For experimental $g_\perp$ of the A$_{1s}$ exciton, we refer to literature values. The experimental values for $g_\parallel$ from this work are shown in the last column. Since in the experiments we can only determine the magnitude but not the sign, we denote only the magnitude $\lvert g_\parallel \rvert$

[a]refs. 28, 61, 67–73
[b]refs. 25, 28, 74
[c]refs. 28, 32, 67, 75–78
[d]ref. 25

their magnitudes are smaller than the experimental values, which are close to reported out-of-plane $g$ factors, i.e., $\lvert g_\parallel(\text{exp.}) \rvert \sim \lvert g_\perp(\text{exp.}) \rvert$. Interestingly, because of the particular symmetry of the bands (CB ~ $\Gamma_9$ and VB ~ $\Gamma_7$ in the $D_{3h}$ point group of the K valleys), only the valence band shows a nonzero value of $g_\parallel$, while for the conduction band it is strictly zero, i.e., the orbital, $L_x^{CB}$, and the spin, $S_x^{CB}$, angular momenta are both zero (cf. table 1). This situation is similar to the zero $g_x$ (Voigt geometry) of the heavy-hole valence band in wurtzite materials with hexagonal symmetry[48–51]. While in the first-principles calculations the interlayer hybridization of electronic bands is fully taken into account, excitonic correlations are not considered. Since we observe the oscillations at excitonic resonances, it is likely that additional hybridization on the excitonic level contribute to the observed $g$ factor. For instance, in-plane magnetic fields introduce a mixing of bright and dark excitons in monolayer TMDCs[30–32]. In the bulk case, excitonic correlations may facilitate the mixing of different exciton channels due to the additional degeneracy of the bands. We emphasize that investigations of these excitonic correlations in the bulk case are beyond the scope of our current study but remain an open topic for future investigations. Also the experimental observation that for the multilayers the $\lvert g_\parallel \rvert$ of the 2s excitons are slightly larger than those of the 1s excitons (cf. Fig. 3a) may be explained by exciton hybridization: The Bohr radius, i.e., the spatial expansion of the 2s excitons is larger than that of the 1s excitons (see Fig. 1e). Therefore, it is likely that hybridization effects may be slightly more important for 2s than for 1s excitons.

In Fig. 3b, a schematic picture of the excitonic center-of-mass dispersion is shown for the relevant excitonic resonances in the two multilayer materials, namely the A$_{1s}$ and A$_{2s}$ resonances. For the excitons, electron and hole spins, $S_z$, are depicted by single-line and double-line arrows, respectively. The energetic splittings, corresponding to the nonzero $g_\parallel$, are taken into account by dashed and solid lines for the center-of-mass parabolas. The helicities are given next to the spin configurations of the excitons. The bold red double arrows should symbolize the coherent oscillations between the excitonic states, when resonantly excited. Hence, we suggest that the observed oscillations originate from coherent oscillations between excitonic levels with different pseudospins, i.e., pseudospin quantum beats. These can be visualized on a Bloch sphere, as shown in Fig. 3c: The north pole corresponds to excitons with pseudospin $\tau = +1$. This means, they occupy the K$^+$ valleys of the individual layers (cf. Fig. 1d). Once they are initialized by a $\sigma^+$ pump pulse, they can coherently oscillate to the south pole, which are excitons with pseudospin $\tau = -1$, i.e., which occupy the K$^-$ valleys of the individual layers. A question, which we can not answer conclusively so far is, if the coherent oscillations are either spin quantum beats of K$^+$ and K$^-$ A excitons solely within the layers (intralayer oscillations), or between the layers (interlayer oscillations), or, a mixture of both. The experimental finding that we do not observe oscillations for the monolayers may favor the scenario of interlayer spin quantum beats in the multilayer samples. This is, furthermore, corroborated by the fact that the interlayer

component of the oscillations is momentum-allowed, since, in $k$ space K$^+$ and K$^-$ valleys are on top of each other in an H-type structure (cf. Fig. 1d). However, it was previously suggested for WSe$_2$ bilayers that only holes may exhibit coherent oscillations in in-plane magnetic fields[9]. Presumably, there may be contributions from both, intralayer- and interlayer oscillations. Which part dominates, we can not say so far. In future investigations this may be further highlighted by experiments on R-type multilayer samples: In contrast to H-type, in R-type stacking, interlayer oscillations of A excitons are momentum forbidden. This scenario may favor intralayer oscillations. However, such experiments will be technically demanding, since the TMDC selenides do not grow in R-type, so, multilayer samples will have to be fabricated manually.

## Discussion

Finally, we would like to make some notes on the layer number dependence. In principle, we would expect the pseudospin oscillations to occur, starting with symmetric H-type bilayer samples, where the spin degeneracy is restored. To elucidate this in more detail, we have computed the $g$ factors for a symmetric WSe$_2$ bilayer (see table SI in the Supplementary Information). We receive indeed for the bilayer a non-zero $g_\parallel$, which is in between the values of the monolayer (where $g_\parallel \sim 0$) and the bulk limit. Also, $g_\perp$ of the bilayer is in between the corresponding values for the monolayer and multilayer (cf. Table SI). Unfortunately, preliminary TRFE experiments on a large-area encapsulated H-type WSe$_2$ bilayer do not show oscillations for an in-plane magnetic field. These preliminary experiments are shown in Fig. S6 of the Supplementary Information, where they are compared to TRFE traces of a closeby multilayer. We speculate that within our laser spot with diameter of about 50 μm on the large-area sample, there may be a large number of microscopic regions with different asymmetric potentials, caused by locally varying strain, dielectric environment, etc., due to the hBN encapsulation, where the spin degeneracy is not restored. This could hinder the development of pseudospin rotations on a large scale. For future experiments, it would be highly desirable to systematically study series of samples with increasing layer number, starting from the bilayer, possibly with smaller laser-spot sizes.

In summary, we have detected ultrafast pseudospin rotations in the GHz to THz frequency range in TMDC multilayers in in-plane magnetic fields via time-resolved Faraday ellipticity. Surprisingly, the magnitudes of the extracted in-plane $g$ factors are close to reported values of out-of-plane $g$ factors of the same materials. This is in stark contrast to monolayer samples, which show no temporal oscillations for nonzero in-plane magnetic field, and which, hence, have an in-plane exciton $g$ factor close to zero. The experimental results are confirmed by first-principles calculations of the $g$ factors. Our study opens the door for manipulation of these pseudospins on ultrafast time scales, making TMDC multilayers an interesting platform for pseudospin operations, possibly putting quantum-gate operations, as suggested in ref. 9, into reach.

## Methods

### Samples

All investigated TMDC samples are mechanically exfoliated from bulk source material (purchased from HQ Graphene) using nitto tape, and then transferred onto transparent sapphire substrates by viscoelastic polymethyldisiloxane stamps[52]. Large-area MoSe$_2$ and WSe$_2$ monolayers are prepared and encapsulated in hexagonal Boron nitride (hBN) multilayers for protection against environmental influences. In the main body of the manuscript, results from two multilayer samples are presented: A WSe$_2$ multilayer, consisting of 14 layers, and a MoSe$_2$ multilayer with about 84 layers.

### Optical experiments

For sample characterization, reflectance-contrast (RC) measurements of all samples are conducted in an optical microscope setup. The samples are mounted by an elastic organic glue on the cold finger of a He-flow cryostat and are kept in vacuum, while the sample holder is cooled down to nominally 5 K. The temperature at the sample position is estimated by the relative intensities of Ruby lines of the sapphire substrate. The substrate temperature is typically between about $T = 10$ K and 30 K. For the RC measurements, a white-light source is used, which is focused by a x60 microscope objective to a spot with diameter of about 10 μm. Reference spectra are recorded at positions next to the TMDC sample. Evaluation of the RC spectra, using a transfer-matrix model, can be found in the Supplementary Information (supplementary Fig. S5)

A schematic picture of the experimental setup, used for TRFE experiments, is shown in supplemental Fig. S1. For TRFE experiments, a mode-locked Ti:Sapphire laser is used, which produces laser pulses with a temporal length of about 80 fs at a repetition rate of 80 MHz. The laser beam is divided into two pulse trains by a beam splitter. The time delay, $\Delta t$, between pump and probe pulses is adjusted by a retroreflector, which is mounted on a linear stepper stage. Both beams are focused by a plano convex lens onto the sample surface, where they overlap. The laser spot diameter at the sample position is about 50 μm. The sample is mounted in an optical cryostat with superconducting magnet coils (split-coil cryostat) at a temperature of about $T = 5$ K, which is maintained by a constant flow of cold He gas. By measuring the laser pulse length before and after the magnet cryostat, we estimate the pulse length at the sample position to be about 130 fs. The pump pulses are circularly polarized and the laser wavelength is tuned to excitonic absorption lines to create a valley polarization in the sample. The temporal dynamics of the valley polarization is then measured by detecting the ellipticity of the linearly-polarized probe pulses after transmission of the sample. For measurement of the ellipticity, a combination of a Wollaston prism, quarter-wave plate and two balanced photo diodes is used. The pump beam is mechanically chopped at a frequency of about 1.6 kHz, and for detection of the photodiode difference signal, lockin technique is used.

### Exciton densities

To get the most accurate estimate of the exciton densities in the experiments, we measure the power of the transmitted pump laser beam for the two cases, when (i) the pump beam is focused on the sample, and (ii) focused next to the sample on the sapphire substrate. The difference in power is the upper limit of the power absorbed by the sample, since with this approach we neglect the difference in reflectivity of the sapphire substrate versus sapphire substrate with TMDC sample. We then assume that the exciton density $n$ is equal to the density of absorbed photons $n_{photons}$, which is related to the absorbed power $P_{abs}$ by $P_{abs} = n_{photons} E_{Laser} f r^2 \pi$. $E_{Laser}$ is the energy of the laser photons, $f$ the repetition rate (80 MHz) of the laser, and $r = 25$ μm the laser-spot radius on the sample.

With this procedure, we get for the WSe$_2$ monolayer an initial exciton density of $n \sim 1.3 \times 10^{12}$ cm$^{-2}$ and for the MoSe$_2$ monolayer $n \sim 1.9 \times 10^{12}$ cm$^{-2}$, when in both cases the A$_{1s}$ exciton is excited resonantly. Both values are well below the Mott density[40,53,54]. For the multilayer samples, we devide the total exciton density by the number of layers to get an estimate of the density per layer. We get for the WSe$_2$ multilayer (14 layers) $n \sim 3.7 \times 10^{11}$ cm$^{-2}$/layer when exciting the A$_{1s}$ exciton resonantly, and, $n \sim 2.1 \times 10^{11}$ cm$^{-2}$/layer for resonant excitation at the A$_{2s}$ exciton. For the MoSe$_2$ multilayer (80 layers), we have $n \sim 1.0 \times 10^{12}$ cm$^{-2}$/layer for the A$_{1s}$ exciton resonance, and $n \sim 1.4 \times 10^{12}$ cm$^{-2}$/layer for the A$_{2s}$ exciton resonance.

### Theoretical modeling

The first-principles calculations are performed within the density functional theory (DFT) using the full-potential all-electron code WIEN2k[55]. We use the Perdew-Burke-Ernzerhof (PBE) exchange-correlation functional[56], a core-valence separation energy of $-6$ Ry, atomic spheres with orbital quantum numbers up to 10 and the plane-wave cutoff multiplied by the smallest atomic radii is set to 9. For the inclusion of spin-orbit coupling, core electrons are considered fully relativistically whereas valence electrons are treated in a second variational step[57]. We use a Monkhorst-Pack k-grid of $15 \times 15 \times 6$ ($15 \times 15$) for the bulk (monolayer). The bulk calculations include van der Waals interactions via the D3 correction[58]. Self-consistency convergence was achived using the criteria of $10^{-6}$ e for the charge and $10^{-6}$ Ry for the energy. The bulk lattice parameters, taken from ref. [59], are $a = 3.282$ Å, $d = 3.340$ Å and $c = 12.960$ Å for WSe$_2$; and $a = 3.289$ Å, $d = 3.335$ Å and $c = 12.927$ Å for MoSe$_2$. Here, the in-plane lattice parameter, $a$, and the layer thickness, $d$, are considered the same for bulk and monolayers. In monolayers, we used a vacuum spacing of 16 Å to avoid interaction among the periodic replicas whereas in the bulk case the total size of the unit cell is the lattice parameter $c$. The calculations of the orbital angular momenta $L_x$ and $L_z$ are based on the fully converged summation-over-bands approach discussed in refs. [47,60–62].

For the calculations of the bulk intralayer excitons we used the effective Bethe-Salpeter equation[63,64]. The energy band dispersion near the K valley is treated as $E(k_x, k_y, k_z) = \frac{\hbar^2}{2m^*}\left(k_x^2 + k_y^2\right) + f(k_z)$, with $m^*$ being the in-plane effective mass and $f(k_z)$ models the dispersion from along the $-H-K-H$ direction of the bulk first Brillouin zone. The DFT calculated in-plane effective masses for WSe$_2$ are $m_{CB} = 0.29 m_0$ and $m_{VB} = 0.36 m_0$, and for MoSe$_2$, $m_{CB} = 0.90 m_0$ and $m_{VB} = 0.61 m_0$. For the function $f(k_z)$, we take the numerical values directly from the DFT calculations. The electron-hole interaction is mediated by the anisotropic Coulomb potential, with the dielectric constants for WSe$_2$ given by $\varepsilon_{xx} = \varepsilon_{yy} = 15.75$ and $\varepsilon_{zz} = 7.75$, and for MoSe$_2$, $\varepsilon_{xx} = \varepsilon_{yy} = 17.45$ and $\varepsilon_{zz} = 8.3$, taken from ref. [65]. Our calculations reveal binding energies of 29.9 (9.1) meV for the $A_{1s(2s)}$ exciton in WSe$_2$ and 41.9 (10.1)meV for the $A_{1s(2s)}$ exciton in MoSe$_2$, respectively. Further details on this approach for intralayer excitons in bulk TMDCs can be found in ref. [66] for bulk WS$_2$.

## Data availability

The data that support the findings within this paper are available within the article and the Supplementary Information file, or available from the corresponding author upon request. Source data of the figures in the paper and the Supplementary Information are provided with this paper as supplementary source data files. Source data are provided with this paper.

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

## Acknowledgements
We gratefully acknowledge valuable discussions with Tobias Korn and would like to thank him for expert help in the initial setup of the experiment. We express our gratitude to Alexey Chernikov for providing the transfer-matrix program, and to Sebastian Bange for expert help with the AFM experiments. Funding by the Deutsche Forschungsgemeinschaft (DFG, German Research Foundation) - Project-ID 314695032 - SFB 1277 (subprojects B05 (C.S.), B07 and B11 (J.F.)), and projects SCHU1171/8-1 (C.S.) and SCHU1171/10-1 (SPP 2244) (C.S.) is gratefully acknowledged. K.W. and T.T. acknowledge support from JSPS KAKENHI (Grant Numbers 19H05790, 20H00354 and 21H05233).

## Author contributions
S.R., D.F., S.F. and P.M. prepared the samples, performed the experiments and analyzed the data. P.E.F.J. and J.F. performed the first-principles and exciton calculations. K.W. and T.T. supported the high-quality hBN material. C.S. conceived the project, analyzed the data and wrote the manuscript. All authors contributed to the discussion of results and to the finalization of the manuscript.

## Funding

## Competing interests
The authors declare no competing interests.
