## [Peer Review File · Nature Communications]

Reviewers' Comments:

Reviewer #1:

Remarks to the Author:

I read this manuscript with great interest. Authors discovered a new effect, i.e., oscillations of the exciton's chiral optical signal in transverse field in a set of multilayer TMD materials. This effect is not expected because in monolayer TMDs transverse field competes with very strong spin orbit coupling that locks such oscillations. Usually, what is observed is just an increase of the relaxation rate but not oscillations. The theoretical explanation of the found effect is still fuzzy but at this stage I find it appropriate. I suppose this kind of results is sufficiently important to be reported in Nature Commun.

The article is very carefully written and well illustrated. So, I congratulate the authors with a great achievement and I recommend to publish the article after the authors consider my optional comments.

1) There are too many references to the obtained results as "cute", "remarkable" etc. throughout the text, and even abstract. While I share some of the excitement, in a serious publication this sounds like the authors are desperate. I suggest to remove some of such words or replace them with something more informative.

2) I think the article would benefit if the authors discuss a path forward for this research. What this effect can be useful for in the future?

Reviewer #2:

Remarks to the Author:

The manuscript "Ultrafast pseudospin quantum beats in multilayer WSe₂ and MoSe₂", written by Simon Raiber and co-authors is devoted to the investigations of excitonic transitions in mono- and multilayer transition metal dichalcogenides (TMDC) by time resolved Faraday ellipticity with in plane magnetic fields. Authors performed both careful measurements and first principles calculations. The presence of non-zero g_{\perp} was revealed. Obtained results allowed to demonstrate ultrafast pseudospin rotations in the GHz- and THz frequency range. Materials under investigation are widely studied nowadays, moreover spin and pseudospin dynamics in TMDC is a vital topic. While the theme of the investigations presented in the manuscript is interesting and important, I have several questions/comments presented below:

1. I suggest to extend the Introduction section and to discuss briefly excitons diffusion in TMDC, as diffusion strongly influence optical and electronic properties of such materials. See, for example, Applied Physics Letters, 2018, 113, 252101, DOI <https://doi.org/10.1063/1.5063263>; Physical Review Letters, 2021, 127, 076801 DOI: <https://doi.org/10.1103/PhysRevLett.127.076801>; Physical Chemistry Chemical Physics 2022 Advance Article DOI <https://doi.org/10.1039/D2CP00557C>.

2. Measurements were performed for the substrate temperature 20 K and for the MoSe₂ multilayer sample it was less than 10K. What was the reason for temperature changing. Could authors comment on the influence of substrate temperature on the obtained results.

3. When discussing biexponential decay in WSe₂ in zero magnetic field under resonant excitation of the A_{1s} exciton authors analyzed several processes which could contribute to the fast decay at short times, but they do not discuss in details mechanisms corresponding to the slow decay component. Can it be due to the presence of defects and traps? Such mechanism were considered in nanoplatelets or perovskites (Nature Communications, 2019, 10, 1–6; Nature communications, 2020, 11, 1–8; Phys.Chem.Chem.Phys., 2020, 22, 24686).

4. Concerning the calculation part, I would like the authors to comment on how could the presence of exciton-exciton interaction influence the obtained results. Does it play any role in the considered experimental conditions?

5. I also would like the authors to comment on the influence of samples heating during the measurements. Could it lead to the shift of the excitonic resonances.

Reviewer #3:

Remarks to the Author:

Here the authors present some of the first experimental studies on in-plane g factor of transition metal dichalcogenides. Notably, with in-plane magnetic field clear oscillations are observed in the TRFE traces of both multilayer WSe₂ and multilayer MoSe₂ and are attributed to coherent oscillations of spin quantum beats. These observations are the first report of such phenomenon in TMDs, provide insight into fundamental properties of TMD systems, and are suitable for publication in Nature Communications following revisions.

As the authors note, they cannot conclusively state whether intralayer or interlayer oscillations are leading to the effect, although they appear to be leaning towards interlayer spin quantum beats. It would be helpful to discuss what future experiments/calculations could aid in the identification of the source of the oscillations.

The authors present data from one WSe₂ multilayer (~14 layers) and one MoSe₂ multilayer (~84 layers). Do the authors expect quantum beats to be present in any multilayer (composed of 2 or more layers), or is this phenomenon only present in thicker, bulk-like samples? Additional data for thinner samples would be beneficial. Are any layer-dependent effects expected as the layer number is decreased?

Do the authors expect quantum beats in multilayered samples that have different structure (i.e R-type) and can such samples be measured?

Dear Dr. Silvia Milana:

Thank you very much for considering our manuscript for publication in Nature Communications and for sending the reports. We would also like to thank the Reviewers for the professional and fair evaluation of our manuscript, and for providing very helpful suggestions and comments, which helped us to improve the manuscript.

Please find in the following our point to point response to the Reviewer comments, together with the changes in the revised manuscript. In the following, Reviewer comments are printed in blue, our replies in black, and changes in the manuscript are indicated by red color.

Reviewer #1:

I read this manuscript with great interest. Authors discovered a new effect, i.e., oscillations of the exciton's chiral optical signal in transverse field in a set of multilayer TMD materials. This effect is not expected because in monolayer TMDs transverse field competes with very strong spin orbit coupling that locks such oscillations. Usually, what is observed is just an increase of the relaxation rate but not oscillations. The theoretical explanation of the found effect is still fuzzy but at this stage I find it appropriate. I suppose this kind of results is sufficiently important to be reported in Nature Commun.

The article is very carefully written and well illustrated. So, I congratulate the authors with a great achievement and I recommend to publish the article after the authors consider my optional comments.

Reply: We sincerely thank the Reviewer for the very kind and encouraging statements.

1) There are too many references to the obtained results as "cute", "remarkable" etc. throughout the text, and even abstract. While I share some of the excitement, in a serious publication this sounds like the authors are desperate. I suggest to remove some of such words or replace them with something more informative.

Reply: We apologize, if we have expressed too much of our excitement about the results in the text. We have done the following modifications:

In the Abstract: "... In ~~stark~~ contrast, we observe ...", "... Our first principles calculations ~~nicely~~ confirm ..."

Page 1, left column: "... Another ~~superior quality property~~ of monolayer material ..."

Page 2, left column: "... While we do not observe a significant influence of in-plane fields of up to 9 T in experiments on monolayers, ~~there is a dramatic effect in multilayer samples: Pronounced pronounced temporal oscillations are observed in the TRFE time traces of multilayers for ...~~"

Page 3, left column: "... decreases ~~dramatically~~ because of the stronger dielectric screening."

Page 3, right column: "... they are ~~drastically significantly~~ different for the multilayer samples."

Page 7, left column: "... The experimental results are ~~nicely~~ confirmed by first-principles calculations ..."

2) I think the article would benefit if the authors discuss a path forward for this research. What this effect can be useful for in the future?

Reply: Thank you very much for this suggestion. In combination with a suggestion of Reviewer #2, we have indicated a possible route for further investigations, starting at the end of the right column on page 6:

"... Presumably, there may be contributions from both, intralayer- and interlayer oscillations. Which part dominates, we can not say so far. In future investigations this may be further highlighted by experiments on R-type multilayer samples: In contrast to H-type, in R-type stacking, interlayer oscillations of A excitons are momentum forbidden. This scenario may favor intralayer oscillations. However, such experiments will be technically demanding, since the TMDC selenides do not grow in R-type, so, multilayer samples will have to be fabricated manually."

We kindly note that in the manuscript, we had already pointed out one possible application of the observed effect for quantum gate operations, as suggested in Ref. 9. In this reference, pseudospin rotations in external electrical and magnetic fields are theoretically investigated with the aim of the realization of quantum gates. To emphasize this point somewhat more, we have changed the text in the left column of page 7 accordingly:

"... Our study opens the door for manipulation of these pseudospins on ultrafast time scales, making TMDC multilayers an interesting platform for pseudospin operations, possibly putting quantum-gate operations, as suggested in Ref. [9], into reach."

Reviewer #2:

The manuscript "Ultrafast pseudospin quantum beats in multilayer WSe₂ and MoSe₂", written by Simon Raiber and co-authors is devoted to the investigations of excitonic transitions in mono- and multilayer transition metal dichalcogenides (TMDC) by time resolved Faraday ellipticity with in plane magnetic fields. Authors performed both careful measurements and first principles calculations. The presence of non-zero g_{\perp} was revealed. Obtained results allowed to demonstrate ultrafast pseudospin rotations in the GHz- and THz frequency range. Materials under investigation are widely studied nowadays, moreover spin and pseudospin dynamics in TMDC is a vital topic. While the theme of the investigations presented in the manuscript is interesting and important, I have several questions/comments presented below:

Reply: We would like to thank the Reviewer for pointing out the importance of our results.

1. I suggest to extend the Introduction section and to discuss briefly excitons diffusion in TMDC, as diffusion strongly influence optical and electronic properties of such materials. See, for example, Applied Physics Letters, 2018, 113, 252101, DOI <https://doi.org/10.1063/1.5063263>; Physical Review Letters, 2021, 127, 076801 DOI:<https://doi.org/10.1103/PhysRevLett.127.076801>; Physical Chemistry Chemical Physics 2022 Advance Article DOI <https://doi.org/10.1039/D2CP00557C>.

Reply: Thank you for pointing out this shortcoming of the introduction. We have now included the 3 mentioned references on exciton diffusion in monolayers as new Refs. 5, 6 and 7, and have inserted in the introduction the sentence: *"... Furthermore, anomalous, nonclassical diffusion behavior of excitons has been detected [5, 6] and calculated [7] for TMDC monolayers."*

2. Measurements were performed for the substrate temperature 20 K and for the MoSe₂ multilayer sample it was less than 10K. What was the reason for temperature changing.

Reply: We apologize that the temperatures in our measurements were not given clear enough in the manuscript to avoid confusion. All TRFE measurements were taken in a split-coil magnet cryostat with variable temperature insert, where the samples were cooled to ~5 K in the flow of cold He gas. This was described in the methods section of the original manuscript on page 7, left column, when the TRFE experiments are described "*... The sample is mounted in an optical cryostat with superconducting magnet coils (split-coil cryostat) at a temperature of about $T = 5$ K, which is maintained by a constant flow of cold He gas. ...*"

To further emphasize this also in the main text, we have added in the figure caption of Fig. 1:

"... We note that the TRFE experiments are all performed at $T \sim 5$ K. ..."

For the reflection contrast experiments, the samples are glued in vacuum on the cold finger of a microscope cryostat. There, the actual sample temperature is not as clear, since the sample is cooled only via the thermal contact of the glue and sapphire substrate, and the temperature may crucially depend on the thermal contact between sample and cold finger. This can lead to slightly different sample temperatures for different samples. Therefore, we took the intensity ratio of two Ruby lines from the sapphire substrates as indication of the sample temperature in these measurements. In the original manuscript, this was explained in the methods section, page 7, left column:

"... For sample characterization, reflectance-contrast (RC) measurements of all samples are conducted in an optical microscope setup. The samples are mounted by an elastic organic glue on the cold finger of a He-flow cryostat and are kept in vacuum, while the sample holder is cooled down to nominally 5 K. The temperature at the sample position is estimated by the relative intensities of Ruby lines of the sapphire substrate. The substrate temperature is typically between about $T = 10$ K and 30 K. ..."

To make this also clearer in the main text, and to point out that different temperatures appeared only in the RC characterization experiments, we have adjusted the corresponding sentence in the figure caption of Fig. 1:

"... The substrate temperature in all RC measurements was $T \sim 20$ K, except for the MoSe₂ multilayer, where it was < 10 K, as derived from the intensity ratio of ruby lines from the substrate (see methods section). ..."

Could authors comment on the influence of substrate temperature on the obtained results.

Reply: Thank you for this important comment, which helped us to evaluate the temperature stability more deeply. From the redshift of the resonance position of the WSe₂ multilayer, we estimate that the observed effect is stable up to ~100 K. We have now elaborated on this on page 4, left column:

"... We note that the A_{1s} resonance position is shifted by about 16 meV to lower energies in comparison to the white-light RC measurements in Fig. 1a, which can be due to bandgap-renormalization effects [39] and/or a temperature increase under pulsed excitation. If the redshift would be caused entirely by a temperature increase, the temperature in the TRFE experiments on

the WSe2 multilayer could be up to $T \sim 100$ K [40] as an upper limit. For the MoSe2 multilayer we get similar results (not shown), though with a resonance maximum much closer to the RC data, most likely due to the much thicker MoSe2 multilayer (84 vs. 14 layers), with much less absorbed power per layer. ..."

Ref. 41 has been added, for the temperature dependence of the effective bandgap of WSe2 bulk material.

3. When discussing biexponential decay in WSe2 in zero magnetic field under resonant excitation of the A1s exciton authors analyzed several processes which could contribute to the fast decay at short times, but they do not discuss in details mechanisms corresponding to the slow decay component. Can it be due to the presence of defects and traps? Such mechanism were considered in nanoplatelets or perovskites (Nature Communications, 2019, 10, 1–6; Nature communications, 2020, 11, 1–8; Phys.Chem.Chem.Phys., 2020, 22, 24686).

Reply: We apologize that we have mentioned and discussed the valley depolarization behavior of monolayer TMDCs only very briefly, since this is not the focus of the present work. We thank the reviewer for the interesting suggestion about results in nanoplatelets, where exciton localization in traps and diffusion contributes to their lifetime, as shown in the last of the 3 references, mentioned above by the reviewer. In TRFE experiments, we measure the time dependence of the valley polarization. This is only true if the exciton lifetime is longer than the valley-polarization lifetime. This is for our monolayer sample indeed the case, we have measured the exciton lifetimes via pump-probe experiments, and it is on the order of 20 ps. The longer component of the TRFE decay of the WSe2 monolayer has a decay constant of 7 ps. Hence, this time is governed by valley depolarization. For the valley depolarization mechanism, we have referred in the manuscript to the established interpretation in literature for clean TMDCs: the intervalley exchange mechanism. We can indeed not completely exclude that exciton localization and diffusion may contribute to the exciton lifetime. Therefore, we have included this now in the revised manuscript on page 3, right column, referring to the Phys. Chem. Chem. Phys. reference:

*"... A significant part of the excitonic population is, however, scattered out of the light cone, e.g., by phonons, and contributes to the valley polarization over a longer time period. **We note that also exciton localization in traps and diffusion, as, e.g., observed for semiconductor nanoplatelets [35], may contribute to a prolonged exciton lifetime.** The main mechanism leading to valley relaxation in WSe2 monolayers, is the long-range exchange mechanism between electron and hole, **which is proportional to the center-of-mass momentum of the exciton [36-38].** The valley-polarization decay time of $\tau_v \sim 7.0$ ps, extracted from the TRFE traces of the hBN encapsulated WSe2 monolayer in Fig. 1f, is in very good agreement with the reported decay time of 6.0 ps, measured on a bare WSe2 monolayer on a SiO2 substrate in Ref. [36], **and with calculations, based on the long-range exchange mechanism [37]. ..."***

4. Concerning the calculation part, I would like the authors to comment on how could the presence of exciton-exciton interaction influence the obtained results. Does it play any role in the considered experimental conditions?

Reply: We thank the referee for this question and apologize for the brief mention of the role of exciton-exciton interaction in the main text, in which we have used the term excitonic correlations. In monolayer TMDCs, in-plane magnetic fields do not introduce a Zeeman splitting for the excitons ($|g| = 0$), but rather mix the bright and dark excitonic states [Refs. 30-32 of the manuscript]. This is the reason why no oscillations are observed in the monolayer limit. For bulk, on the other hand, our first principles calculations firmly establish that there is a Zeeman splitting contribution for the excitons (the nonzero in-plane g factors, primarily originated from the interlayer coupling in the valence band). Due to the additional degeneracy of the bulk bands, excitonic correlations may also introduce a mixing of different exciton species, thus renormalizing the final observed g factor. This could be one of the reasons for the small discrepancy we observe between our calculated and measured g -factor values. Certainly a deeper theoretical investigation of these excitonic correlations, within the robust formalism of the full ab initio GW-BSE for example, would be very valuable, but we believe this is beyond the scope of our current work, since the main physical mechanism behind the oscillation is the emergence of the non-zero g factors for in-plane magnetic fields.

Considering the question from the referee, we modified the text of the manuscript on page 5, right column, as follows:

"... it is likely that additional hybridization on the excitonic level contribute to the observed g factor. For instance, in-plane magnetic fields introduce a mixing of bright and dark excitons in monolayer TMDCs [30-32]. In the bulk case, excitonic correlations may facilitate the mixing of different exciton channels due to the additional degeneracy of the bands. We emphasize that investigations of these excitonic correlations in the bulk case are beyond the scope of our current study but remain an open topic for future investigations. Also the observation that for the multilayers the $|g|$ of the 2s excitons are slightly larger than those of the 1s excitons (cf. Fig. 3a) ~~points into this direction~~ may be explained by exciton hybridization: The Bohr radius, i.e., the spatial expansion of the 2s excitons is larger than that of the 1s excitons (see Fig. 1e). Therefore, it is likely that hybridization effects may be slightly more important for 2s than for 1s excitons...."

5. I also would like the authors to comment on the influence of samples heating during the measurements. Could it lead to the shift of the excitonic resonances.

Reply: Please see answer to the question about the influence of the sample temperature, above. There, we elaborate on the observed shift of the excitonic resonances.

Reviewer #3:

Here the authors present some of the first experimental studies on in-plane g factor of transition metal dichalcogenides. Notably, with in-plane magnetic field clear oscillations are observed in the TRFE traces of both multilayer WSe₂ and multilayer MoSe₂ and are attributed to coherent oscillations of spin quantum beats. These observations are the first report of such phenomenon in TMDs, provide insight into fundamental properties of TMD systems, and are suitable for publication in Nature Communications following revisions.

Reply: Thank you very much for the very positive statements.

As the authors note, they cannot conclusively state whether intralayer or interlayer oscillations are leading to the effect, although they appear to be leaning towards interlayer spin quantum beats. It would be helpful to discuss what future experiments/calculations could aid in the identification of the source of the oscillations.

Reply: Thank you for this constructive comment. In combination with your interesting suggestion about experiments in R-type samples, below, this has led us to an important suggestion, which is now mentioned on page 6, right column:

"... In future investigations this may be further highlighted by experiments on R-type multilayer samples: In contrast to H-type, in R-type stacking, interlayer oscillations of A excitons are momentum forbidden. This scenario may favor intralayer oscillations. However, such experiments will be technically demanding, since the TMDC selenides do not grow in R-type, so, multilayer samples will have to be fabricated manually. ..."

The authors present data from one WSe₂ multilayer (~14 layers) and one MoSe₂ multilayer (~84 layers). Do the authors expect quantum beats to be present in any multilayer (composed of 2 or more layers), or is this phenomenon only present in thicker, bulk-like samples? Additional data for thinner samples would be beneficial. Are any layer-dependent effects expected as the layer number is decreased?

Reply: This is of course a very important and absolutely valid question, on which we definitely should comment on. In principle, we would expect oscillations to occur, starting from a bilayer, since there, spin degeneracy is restored. The comment motivated us to perform additional calculations and to make additional experiments. We have performed calculations on a symmetric WSe₂ bilayer and included them in the supplementary information. The calculated g factors are indeed in between the values of the monolayer and the multilayer. To test this experimentally, we have successfully prepared an hBN-encapsulated, large-area WSe₂ bilayer sample, which is next to a thick (~100 layers) bulk part. A microscope image of the sample is shown in the new supplementary Fig. S6a. For convenience, we have reproduced Fig. S6 below, on the next page. The two dashed circles mark approximately the positions and sizes of the laser spots, used for TRFE experiments on the bilayer (blue circle) and the multilayer regions (orange circle). For the TRFE experiments, the laser wavelength was tuned to be in resonance with the A_{1s} excitons of the two materials. TRFE traces, measured on these spots for an in-plane magnetic field of $B_{||} = 6$ T are shown in Figs. S6b and S6c for the bilayer and multilayer regions, respectively. While the temporal oscillations of the TRFE signal, known from the multilayer samples in the main part of the manuscript, can be nicely reproduced in the multilayer region, the signal of the bilayer does not show oscillations. As elaborated in the revised manuscript, we would expect oscillations also on the bilayer, though presumably with an approximately by a factor of two larger period, as obtained from the calculations on the bilayer. We can only speculate about the absence of oscillations in the experiment on the bilayer. We would expect oscillations to occur for a symmetric bilayer sample, since there, the spin degeneracy is restored. The large-area encapsulated bilayer sample for sure has local inhomogeneities due to strain or varying dielectric environment because of better or worse local contact between the hBN and the bilayer. So, there may be a locally varying asymmetric

potential, over which we average with our large ($\sim 50 \mu\text{m}$) spot size. We believe that this may be the reason, hindering the appearance of pseudospin oscillations in the TRFE experiment.

Fig. S6: (a) Microscope image of an hBN-encapsulated bilayer WSe₂ sample (upper, gray region) and a closeby bulk part (lower, bright region). The dashed circles mark positions of the laser spot for TRFE experiments in an in-plane magnetic field. (b) and (c) TRFE traces, measured on the sample spots, marked in (a), for an in-plane field of $B_{\parallel} = 6 \text{ T}$. The laser energies are chosen to be in resonance with the A1s exciton in the corresponding sample.

We have included a table (table SI) with the results of the calculations for a WSe₂ bilayer, the new Fig. S6, and some explanatory text in the supplementary information.

In the main part of the manuscript, we have added the following text in the right column on page 6:

"... Finally, we would like to make some notes on the layer number dependence. In principle, we would expect the pseudospin oscillations to occur, starting with symmetric H-type bilayer samples, where the spin degeneracy is restored. To elucidate this in more detail, we have computed the g factors for a symmetric WSe₂ bilayer (see table SI in the supplementary information). We receive indeed for the bilayer a non-zero g_{\parallel} , which is in between the values of the monolayer (where $g_{\parallel} \sim 0$) and the bulk limit. Also, g_{\perp} of the bilayer is in between the corresponding values for the monolayer and multilayer (cf. table SI). Unfortunately, preliminary TRFE experiments on a large-area encapsulated H-type WSe₂ bilayer do not show oscillations in an in-plane magnetic field. These preliminary experiments are shown in Fig. S6 of the supplementary information, where they are compared to TRFE traces of a closeby multilayer. We speculate that within our laser spot with diameter of about $50 \mu\text{m}$ on the large-area sample,

there may be a large number of microscopic regions with different asymmetric potentials, caused by locally varying strain, dielectric environment, etc., due to the hBN encapsulation, where the spin degeneracy is not restored. This could hinder the development of pseudospin rotations on a large scale. For future experiments it would be highly desirable to systematically study series of samples with increasing layer number, starting from the bilayer, possibly with smaller laser-spot sizes. ..."

Do the authors expect quantum beats in multilayered samples that have different structure (i.e R-type) and can such samples be measured?

Reply: Please see our answer to your first comment, above.

Dear Editor, we hope that with the applied changes the revised manuscript is suitable for publication in Nature Communications.

Thank you very much!

With best regards

Christian Schüller

Reviewers' Comments:

Reviewer #2:

Remarks to the Author:

The authors strongly modified the manuscript following my comments and suggestions. The answer is complete and modifications performed in the Manuscript are well done. Wright now, the Manuscript not only contains new physics but is well written and very understandable for the readers. I think in its present form the manuscript can be published in Nature Communications.

Reviewer #3:

Remarks to the Author:

In the revised manuscript and rebuttal letter, the authors have adequately addressed all the questions and concerns raised by the reviewers. I recommend publication in Nature Communications.

Reviewer #2:

The authors strongly modified the manuscript following my comments and suggestions. The answer is complete and modifications performed in the Manuscript are well done. Wright now, the Manuscript not only contains new physics but is well written and very understandable for the readers. I think in its present form the manuscript can be published in Nature Communications.

Reply: We sincerely thank the Reviewer for the positive statements.

Reviewer #3:

In the revised manuscript and rebuttal letter, the authors have adequately addressed all the questions and concerns raised by the reviewers. I recommend publication in Nature Communications.

Reply: We would like to thank the Reviewer for the positive evaluation.

All changes due to the Author Checklist are marked with red color in the manuscript file.

Dear Editor, we hope that with the applied changes the revised manuscript can be finally accepted for publication in Nature Communications.

Thank you very much!

With best regards

Christian Schüller